# Characterization of MSC Growth, Differentiation, and EV Production in CNF Hydrogels Under Static and Dynamic Cultures in Hypoxic and Normoxic Conditions

**DOI:** 10.3390/bioengineering11101050

**Published:** 2024-10-21

**Authors:** Ilias Nikolits, Farhad Chariyev-Prinz, Dominik Egger, Falk Liebner, Nicolas Mytzka, Cornelia Kasper

**Affiliations:** 1Institute of Cell and Tissue Culture Technologies, Department of Biotechnology, University of Natural Resources and Life Sciences Vienna, Muthgasse 18, 1190 Vienna, Austria; ilias.nikolits@boku.ac.at (I.N.); farhad.cp@boku.ac.at (F.C.-P.); 2Institute of Cell Biology and Biophysics, Leibniz University Hannover, Herrenhäuser Strasse 2, 30419 Hannover, Germany; egger@cell.uni-hannover.de; 3Institute of Chemistry of Renewable Resources, Department of Chemistry, University of Natural Resources and Life Sciences Vienna, Konrad Lorenz Strasse 24, 3430 Tulln, Austria; falk.liebner@boku.ac.at; 4MicroDiagnostics Unit, Fraunhofer Institute for Cell Therapy and Immunology, Perlickstraße 1, 04103 Leipzig, Germany; nicolas.mytzka@izi.fraunhofer.de

**Keywords:** mesenchymal stem cells, cellulose, hydrogels, hypoxia, extracellular vesicles, immunomodulation

## Abstract

Mesenchymal stem cells (MSCs) hold immense therapeutic potential due to their regenerative and immunomodulatory properties. However, to utilize this potential, it is crucial to optimize their in vitro cultivation conditions. Three-dimensional (3D) culture methods using cell-laden hydrogels aim to mimic the physiological microenvironment in vitro, thus preserving MSC biological functionalities. Cellulosic hydrogels are particularly promising due to their biocompatibility, sustainability, and tunability in terms of chemical, morphological, and mechanical properties. This study investigated the impact of (1) two physical crosslinking scenarios for hydrogels derived from anionic cellulose nanofibers (*to*-CNF) used to encapsulate adipose-derived MSCs (adMSCs) and (2) physiological culture conditions on the in vitro proliferation, differentiation, and extracellular vesicle (EV) production of these adMSCs. The results revealed that additional Ca^2+^-mediated crosslinking, intended to complement the self-assembly and gelation of aqueous *to*-CNF in the adMSC cultivation medium, adversely affected both the mechanical properties of the hydrogel spheres and the growth of the encapsulated cells. However, cultivation under dynamic and hypoxic conditions significantly improved the proliferation and differentiation of the encapsulated adMSCs. Furthermore, it was demonstrated that the adMSCs in the CNF hydrogel spheres exhibited potential for scalable EV production with potent immunosuppressive capacities in a bioreactor system. These findings underscore the importance of physiological culture conditions and the suitability of cellulosic materials for enhancing the therapeutic potential of MSCs. Overall, this study provides valuable insights for optimizing the in vitro cultivation of MSCs for various applications, including tissue engineering, drug testing, and EV-based therapies.

## 1. Introduction

Mesenchymal stem cells (MSCs) represent a promising source for cell-based therapies, thanks to their multipotent differentiation capacity [1], their immunomodulatory properties [2], and their availability from various tissue sources [3]. In order to establish more physiologically relevant culture conditions and to better preserve the biological functionalities and characteristics of MSCs in vitro, 3D culture methods are utilized to emulate the microenvironment that cells encounter in vivo [4].

Incorporating MSCs into hydrogels constitutes a core strategy for recapitulating the native interactions between cells and the extracellular matrix (ECM), thereby preserving the physiological properties of the cells [5]. Hydrogels are water-swollen natural or synthetic polymeric networks, which are frequently used in research, diagnostics, and therapeutic applications as their adjustable chemical structure, network morphology, and mechanical properties entail biocompatibility and, in many cases, resemblance to the cell ECM structure [5]. Cell culture applications using hydrogels with embedded MSCs encompass tissue engineering [6], drug testing, and disease modeling [7] as well as, more recently, platforms for the production [8] and delivery [9] of extracellular vesicles (EVs) for therapeutic purposes.

Among the various natural biomaterials investigated for hydrogel synthesis, cellulose stands out due to its cost effectiveness, abundance, sustainability, as well as its demonstrated excellent biocompatibility with various human cell types [10]. In contrast to widely utilized animal-derived natural biomaterials, including collagen, gelatin, and fibrin, which are commonly employed in the fabrication of biomaterial scaffolds [11], plant-based biomaterials such as cellulose exhibit reduced immunogenicity [12] and their synthesis and in vitro applications align with the principles of the 3Rs (Replacement, Reduction, and Refinement) in experimentation [13]. Moreover, native cellulose offers facile modification opportunities due to its abundant accessible hydroxyl groups [10].

Hydrogels prepared from anionic cellulose nanofibers (*to*-CNFs)—the latter obtained by cellulose oxidation using the metastable2,2,6,6-tetramethylpiperidine-1-oxyl (TEMPO) radical and subsequent delamination of the cellulose fibers by shear forces—have been demonstrated to support the in vitro cultivation of human breast cancer cells and mouse embryonic stem cells [14]. The authors have recently shown that this type of hydrogel can effectively support the spreading, proliferation, and migration of human adipose-derived primary MSCs (adMSCs) [15]. The authors’ previous investigations on *to*-CNF hydrogels have confirmed that anionic *to*-CNFs, besides their well-known nematic self-assembly in a dilute aqueous dispersion state and gel formation triggered by acid-induced hydrogen bond formation, also interact with divalent cations present in the culture medium, such as Ca^2+^ or Mg^2+^ [16]. Similarly, Ca^2+^-induced ionic crosslinking has been explored in wound healing [17] and osteoblast cultivation [18] in vitro studies to further enhance the stability and integrity of *to*-CNF hydrogels.

In addition to biomimetic materials for 3D cultivation, the optimization of more fundamental in vitro conditions has shown promise in preserving in vivo-like cell behavior. The dynamic cultivation of MSCs has been found to enhance proliferation [19], differentiation [20], and EV production capacities [21] as it mimics the continuous dynamic environment that cells experience in vivo. Another critical physiological parameter is oxygen concentration, which has gained significant scientific interest since the discovery of the hypoxia-inducible (HIF)-1 transcription factor. The commonly used 21% O_2_ tension in stem cell cultures is considered hyperoxic compared to the average of 5% found in stem cell niches in vivo [22]. In vitro hypoxia has been shown to affect numerous MSC features, including metabolism, proliferation, differentiation, immunophenotype expression, migration, apoptosis [23], as well as EV production and functionality [24]. Therefore, the strategic combination of these physiological parameters and methodologies holds significant potential for enhancing the therapeutic and diagnostic capabilities of in vitro cultured MSCs.

EVs are lipid bilayer-enclosed structures containing proteins, lipids, and nucleic acid fragments, which are released by cells into the extracellular environment. They have recently garnered significant interest due to their roles in intercellular communication and potential therapeutic applications [25]. EVs can be broadly classified into three main types based on their size and biogenesis: exosomes (30–150 nm), microvesicles (100–1000 nm), and apoptotic bodies (500–2000 nm). High-impact studies have demonstrated the role of EVs in various physiological and pathological processes, including cancer progression, immune modulation, and tissue regeneration [25,26]. For instance, research has shown that EVs derived from mesenchymal stem cells (MSCs) can promote tissue repair and modulate immune responses, highlighting their potential in regenerative medicine [27]. Additionally, EVs are being explored as biomarkers for disease diagnosis and as vehicles for targeted drug delivery [26].

This study sought to improve the in vitro cultivation of human adMSCs within *to*-CNF hydrogel spheres. To this end, the authors investigated different crosslinking strategies for aqueous *to*-CNF dispersions and, more importantly, assessed the combinatorial effect of physiological conditions on cellular capabilities like proliferation, differentiation, and immunomodulation in vitro. Drawing on the existing literature, the authors explored the effects of an additional Ca^2+^-induced crosslinking, in addition to the previously described nematic self-assembly process [15], focusing on the mechanical properties of the resultant hydrogel spheres and their impact on encapsulated cultured cells. Subsequently, the authors examined the impact of static versus dynamic and normoxic (21% O_2_) versus hypoxic (5% O_2_) cultivation conditions on cell growth and differentiation capacities in response to more physiological cues during in vitro culture. This aspect was particularly intriguing as there is a paucity of studies investigating the effects of physiological in vitro culture aspects using cellulose-based hydrogels and MSCs. Finally, the potential of *to*-CNF hydrogel spheres as platforms for therapeutic EV production from adMSCs in scalable bioreactor systems was evaluated. Cellulose hydrogels have been studied as cargo-loaded carriers for EVs in a limited number of studies [28,29], but, to the authors’ knowledge, not as platforms for the production of EVs from encapsulated MSCs. Given that EV production benefits from the use of bioreactors [21] and large quantities of generated MSC EVs are required for therapeutic doses [30], the authors tested the capacity of adMSCs encapsulated in *to*-CNF hydrogel spheres to produce EVs during cultivation in a scalable vertical wheel bioreactor setup.

The findings of this study further underscore the potential of *to*-CNF hydrogels and physiological culture conditions to benefit various applications including in vitro modeling, drug testing, and EV production.

## 2. Materials and Methods

If not stated otherwise, reagents were purchased from Sigma-Aldrich, Vienna, Austria.

### 2.1. Cells

#### 2.1.1. adMSCs

The use of human tissue was approved by the ethics committee of the University of Lübeck, Germany (AZ 20-333, 4 November 2020), and the donor provided written consent. Human adipose-derived MSCs (adMSCs) were isolated within 24 h after the abdominal plastic surgery of a 29-year-old male donor, as previously described [31]. Briefly, the fat tissue was minced with scissors and digested with collagenase type IA for 1 h at 37 °C. A series of centrifugation and washing steps were performed to obtain the stromal vascular fraction, which was then transferred into cell culture flasks and cultivated in an expansion medium composed of MEM alpha (Minimum Essential Medium; Thermo Fisher Scientific, Waltham, MA, USA), 0.5% gentamicin (Lonza, Basel, Switzerland), 2.5% human platelet lysate (hPL, PL BioScience, Aachen, Germany), and 1 U/mL heparin (PL BioScience, Aachen, Germany) in a humidified incubator at 37 °C and 5% CO_2_. When adMSCs reached approximately 80% confluency, they were detached using accutase (GE Healthcare, Little Chalfont, UK) and cryopreserved in MEM alpha, 2.5% hPL, 10% dimethyl sulfoxide (DMSO), and 1 U/mL heparin in liquid nitrogen. For *to*-CNF hydrogel sphere encapsulation, cells were thawed, expanded for up to three passages in cell culture flasks (Sarstedt, Nümbrecht, Germany), and harvested by accutase treatment.

#### 2.1.2. PBMCs

Whole blood samples from 5 donors were provided by the Red Cross Austria, Vienna (EK20210506_01). Samples were first diluted in a 1:2 ratio with RPMI 1640 culture medium (Thermo Fisher Scientific) supplemented with 2.5% hPL and 1 U/mL heparin and transferred dropwise into tubes containing Ficoll-Paque PLUS (Cytiva, Marlborough, MA, USA) in a ratio of 2:5. Tubes were then centrifuged twice at 500× *g*, first for 20 and then for 15 min. Subsequently, the supernatant plasma layer was discarded, and the peripheral blood mononuclear cell (PBMC) layer was collected. PBMCs were then washed twice with phosphate buffer saline solution (PBS) and centrifuged at 350× *g* for 5 min. PBMCs from different donors were pooled together and stained with 5 μM of carboxyfluorescein succinimidyl ester (CFSE, Merck, Darmstadt, Germany) for 15 min at 37 °C protected from light. Following staining, cells were washed with RPMI 1640 medium at 500 G for 5 min and cryopreserved in RPMI 1640, 2.5% hPL, 10% DMSO, and 1 U/mL of heparin in liquid nitrogen.

### 2.2. Preparation of to-CNFs

The aqueous dispersion of negatively charged *to*-CNFs was prepared following a previously established protocol. At first, 40 g (dry weight) of never-dried bisulfite hardwood dissolving pulp (50% water content) was suspended in 1.8 L of deionized water and disintegrated using a household blender for 1 min. Next, 640 mg of 2,2,6,6-tetramethylpiperidine-1-oxyl (TEMPO) (0.1 mmol/g of cellulose) and 4 g of NaBr (1 mmol/g of cellulose) were added to selectively oxidize the primary hydroxyl groups (C6) of cellulose. Then, 60 mL of NaClO solution (available chlorine 10–15%) was added gradually to the mixture at room temperature, under constant stirring (1000 rpm) and with continuous pH adjustment to 10 accomplished using 0.1 M of NaOH. After the addition of NaClO, the oxidized cellulose was given 30 min of reaction time under constant stirring before it was washed with deionized water and filtered multiple times. The oxidized pulp was then postoxidized by suspending the material in 1 L of 0.1 M CH_3_COOH solution and adding 9.72 g of NaClO_2_ (2.7 mmol/g of cellulose), under constant stirring (1000 rpm) at room temperature overnight, in order to convert the remaining intermediary aldehyde groups into carboxyl moieties. Following chlorite oxidation, and multiple washing steps by vacuum-aided filtration, the obtained 6-carboxyl cellulose was disintegrated in water to give a 0.5% *w*/*w* suspension. After adjusting pH 8 by the addition of diluted NaOH, the aqueous suspension was subjected to repeated (eight passes) mechanical defibrillation at 80 MPa using a benchtop homogenizer (APV 1000, AxFlow GmbH, Premstätten, Austria). Aiming to reduce viscosity (and hence heating during nanofibrillation), the dispersion was diluted to 0.125% during the first three passes. After homogenization, the dispersion of negatively charged cellulose nanofibrils was up-concentrated to a final concentration of 0.7% wt. using a rotary evaporator (40 °C, 4.5 kPa), and the pH was adjusted to 7.

### 2.3. Cell Encapsulation in to-CNF Hydrogels Spheres for In Vitro Cultivation

adMSCs were thawed and expanded for one passage in cell culture flasks, following the procedure detailed in Section 2.1. After expansion, the cells were harvested using accutase and resuspended in an appropriate volume of standard expansion medium (composed of MEM alpha, 2.5% ultracentrifuged human platelet lysate, and 1 U/mL heparin) for encapsulation in the hydrogel spheres. The final aqueous *to*-CNF dispersion (0.7% wt.) was combined with the cell suspension using a Vortex-Genie 2 vortex mixer to achieve a bulk concentration of 0.5% wt. *to*-CNFs in the final hydrogels.

Subsequently, 15 μL of the viscous cell-containing material was pipetted until it formed a hanging drop and placed in a 6-well plate (6 hydrogel spheres per well). The CNF hydrogel spheres were allowed to crosslink for 10 min, with or without being submerged in a 100 mM CaCl_2_ solution in a standard expansion medium (pH 7). The concentration of the CaCl_2_ crosslinking solution was chosen based on parameters established in other relevant studies [17,18,32,33]. After incubation, the samples submerged in the CaCl_2_ solution were washed twice with a standard expansion medium, and then 4 mL of the same medium was added to all wells. The samples were placed in an incubator at 37 °C and 5% CO_2_, with oxygen levels set at either 21% or 5%, and were either kept static or placed on a gyroscopic shaker at 70 rpm during cultivation. The well plates were coated with agarose to prevent background noise in the analyses from cells growing on the tissue culture-treated surface of the wells.

Samples were prepared with a cell concentration of 5 × 10^5^ cells/mL of hydrogel spheres, and hydrogel spheres without cells served as controls. Media changes were performed twice per week.

In order to enhance the physiological relevance of the study’s outcomes, all cell culture experiments were performed without the addition of antibiotic supplements. Although antibiotics offer protective effects, their documented detrimental impact on cell culture warranted their exclusion [34].

### 2.4. to-CNF Hydrogel Sphere Cultivation in Bioreactors

*to*-CNF hydrogel spheres were prepared according to the methodology described in Section 2.3. After preparation, the hydrogel spheres were placed in 6-well plates and cultivated on a gyroscopic shaker at 70 rpm for 24 h prior to bioreactor inoculation. Subsequently, the PBS-0.1 Mini vertical wheel bioreactors (PBS Biotech, Camarillo, CA, USA) were seeded with 60 *to*-CNF hydrogel spheres and 100 mL of standard expansion medium. The mixing speed was set at 30 rpm, and the bioreactors were placed in an incubator maintained at 37 °C, 5% CO_2_, and either 21% or 5% O_2_ for 21 days. Medium replacement was performed once per week, replacing half the volume of the culture.

To eliminate xenogenic components and minimize the contamination of harvested EV populations with non-MSC EVs, ultracentrifuged hPL was used as a serum supplement in the adMSC EV production cultures [35].

### 2.5. Live/Dead Cell Visualization

After the cultivation period, samples were stained with 4 μL/mL of calcein acetoxymethyl (calcein AM; Invitrogen, Thermo Fisher Scientific, Waltham, MA, USA) and 4 μL/mL of propidium iodide (PI; Invitrogen) in PBS to visualize live and dead cells. The samples were then incubated for 1 h on a gyroscopic shaker (70 rpm) at 37 °C in the dark, washed twice with PBS, and transferred to a fresh standard expansion medium for fluorescence microscopic analysis.

### 2.6. Cell Metabolic Activity Assessment

The metabolic activity of the *to*-CNF hydrogel sphere-encapsulated adMSCs was assessed using a resazurin-based viability assay (In vitro Toxicology Assay Kit; TOX8) according to the manufacturer’s instructions. After removing the supernatant from the cultures, the same volume of standard expansion medium containing 10% TOX8 was added. The welled plates were then incubated for 3 h on a gyroscopic shaker (70 rpm) at 37 °C and 5% CO_2_. Following incubation, the fluorescence intensity of the supernatant was measured at 590 nm at an excitation wavelength of 560 nm using an Infinite M1000 plate reader (Tecan, Männedorf, Switzerland). The measured values were normalized to blank controls.

### 2.7. DNA Quantification

At the end of the cultivation period in the well plates, the culture medium was removed, and the *to*-CNF hydrogel sphere samples were washed with PBS. Subsequently, they were individually transferred to 1.5 mL Eppendorf tubes and mechanically disrupted using a spatula. Next, 500 μL of lysis buffer (10 mM of Tris HCl pH 8.5 (AppliChem, Darmstadt, Germany), 5 mM of ethylenediaminetetraacetic acid (EDTA), 0.2% sodium dodecyl sulfate (SDS; AppliChem), 0.2 M of NaCl, and 0.1 mg/mL of proteinase K) was added to each tube and samples were incubated overnight on an Eppendorf Thermomixer comfort (Eppendorf, Hamburg, Germany) at 55 °C and 700 rpm. Following the overnight incubation, the temperature was increased to 85 °C for 45 min for proteinase K inactivation, and then cooled down at 4 °C for 20 min. The samples were centrifuged at 1000× *g* for 2 min and the supernatant was carefully aspirated without disturbing the hydrogel fragment pellet.

The DNA content in the lysed sample supernatants was determined using a Quant-iT™ PicoGreen™ dsDNA Assay Kit (Thermo Fischer Scientific), according to the manufacturer’s instructions. The fluorescence intensity of the samples was measured at 520 nm with an excitation wavelength at 480 nm, using an Infinite M1000 plate reader (Tecan). The measured values were normalized to blank controls and to the DNA content of known cell numbers using the same kit and cells.

### 2.8. Isolation of Extracellular Vesicles

Cell culture supernatants collected during media changes were pooled together and consecutively centrifuged at 300× *g* (10 min) and 2000× *g* (20 min) at room temperature to remove cells and debris. These solutions were then transferred into ultracentrifugation tubes (Beckman-Coulter, Brea, CA, USA) and subjected to an additional centrifugation step at 100,000× *g*, 4 °C, for 90 min using an Optima LE-80 ultracentrifuge with an SW 32 Ti rotor (Beckman Coulter). The resulting pellets were resuspended in 200 μL of Ca^2+^/Mg^2+^-free PBS (PAN Biotech, Aidenbach, Germany) and stored at −80 °C until used for further analysis. The same volume of MEM alpha medium with 2.5% ultracentrifuged hPL and 1 U/mL of heparin was processed accordingly for use as a blank control.

### 2.9. Protein Concentration Assay

The protein concentration of the cell culture media supernatants, following clearing and ultracentrifugation, was measured using a BCA protein Assay Kit (Micro BCA™ Protein Assay Kit, Thermo Fischer Scientific). The absorbance of the samples was measured at 562 nm, using an Infinite M1000 plate reader (Tecan). The measured values were normalized to blank controls.

### 2.10. Nanoparticle Tracking Analysis

The particle concentration and size distribution in the cell culture supernatants were determined using nanoparticle tracking analysis (NTA). To distinguish EVs from non-lipid membrane particles, samples were stained with CellMask™ (Invitrogen). Stained samples were diluted in Ca^2+^/Mg^2+^-free PBS and analyzed using a ZetaView x30 Twin (Particle Metrix, Inning, Germany). Data were acquired in one cycle of measurement over 11 laser scattering and fluorescence measurement positions and analyzed using the corresponding instrument software (ZetaView 8.05.16 SP3, Particle Metrix).

### 2.11. PBMC Co-Culture with adMSCs or adMSC-EVs

The immunomodulatory potential of EVs derived from adMSCs encapsulated within *to*-CNF hydrogel spheres during cultivation in the vertical wheel bioreactor was evaluated using a previously described T-cell-based potency assay [36]. In brief, PBMCs were seeded in 24-well plates at a density of 5 × 10^5^ cells/well in RPMI 1640 medium supplemented with 2.5% hPL and 1 U/mL heparin. Following a 24 h incubation period, either 1 × 10^5^ adMSCs or 4 × 10^8^ adMSC-derived extracellular vesicles (adMSC-EVs) produced from normoxic and hypoxic cultures in a vertical wheel bioreactor were added to each well. In addition, 5 μg/mL of phytohemagglutinin (PHA) was included in the culture medium to stimulate PBMCs. The cells were co-cultured for a total of five days.

After the culture period, non-adherent PBMCs were collected from the supernatant medium and washed with phosphate-buffered saline (PBS) at 500 G for 5 min. The PBMC samples were then stained with a prepared CD3 antibody solution (Thermo Fisher Scientific) at a dilution of 1:200 in PBS, and incubated at 4 °C for 30 min in the dark. Subsequently, the samples were washed once more with PBS at 500 G for 5 min and finally resuspended in PBS supplemented with 2 mM EDTA for flow cytometry analysis. The measured values were normalized to unstimulated controls without PHA.

The concentration of adMSC-EVs added to the PBMC culture was determined based on the reported production rates of EVs per MSC in 3D cultivation systems [37], and was chosen to be in accordance with the reported total number of particles per cell from a different study [38].

### 2.12. Flow Cytometry Analysis

Samples for flow cytometry analysis were processed using a Cytoflex 2L cytometer (Beckman-Coulter) equipped with 488 and 638 nm lasers. Data were collected for 10^5^ events at a flow rate of 30 μL/min and analyzed using Kaluza 2.1 software (Beckman-Coulter).

### 2.13. Extracellular Vesicle Surface Marker Analysis

The analysis of the EV surface markers derived from adMSCs encapsulated within *to*-CNF hydrogel spheres during vertical wheel bioreactor cultivation utilized a microarray-based approach. Initially, 3D-epoxy functionalized polymer slides (PolyAn, Berlin, Germany) were employed for spotting with antibody solutions, each supplemented with 2.5% glycerol. Antibody solutions were individually prepared, each containing either purified anti-human CD9, Ultra-LEAF™ Purified anti-human CD63, Purified anti-human CD81 (TAPA-1), or Purified Mouse IgG1, κ Isotype Ctrl (all Biolegend, San Diego, CA, USA) at a final concentration of 0.1 mg/mL in PBS. Droplets of 2.5 nL of the antibody solutions were dispensed onto the slides using a PDC 70 Piezo Dispensing Capillary (Scienion, Berlin, Germany). Nine spots of each antibody solution were prepared for each sample as replicates. Following spotting, the slides were air-dried for 30 min at room temperature. Subsequently, a 16-well ProPlate^®^ Multi-Array Slide System (Grace Bio-Labs, Bend, OR, USA) was utilized for EV-specific antibody staining on the spotted microarray slides. After assembling the slide system according to the manufacturer’s instructions, each well was filled with 200 μL of PolyAn Blocking Solution (PolyAn) and incubated for 1 h at room temperature with agitation on an orbital shaker (450 rpm). Following incubation, the wells were washed three times with 100 μL of PBS for 5 min under agitation on an orbital shaker (450 rpm). Next, 100 μL of EV samples (containing approximately 3.5 × 10^8^ EVs) were added to each well and incubated for 1 h at room temperature with orbital mixing (450 rpm). Following incubation with the EV samples, the washing step was repeated with PBS. Subsequently, 100 μL of the antibody solution comprising 0.12 μg/mL of APC Mouse Anti-Human CD9, 4 μg/mL of APC Mouse Anti-Human CD63, and 4 μg/mL of APC Mouse Anti-Human CD81 (all Biolegend) in PBS was added to each well and incubated for 1 h at room temperature with agitation (450 rpm). After incubation, the washing step was repeated, and the slide was removed and washed with deionized water before being blow-dried with nitrogen. Finally, the fluorescence intensity of the sample slides was measured at 635 nm using a GenePix Professional 4200 A microarray scanner (Molecular Devices, San Jose, CA, USA).

### 2.14. Adipogenic Differentiation

The differentiation potential of adMSCs encapsulated in *to*-CNF hydrogel spheres was assessed as follows. Samples were prepared according to Section 2.3 and incubated at 37 °C with 5% CO_2_ and either 21% or 5% O_2_ on a gyroscopic shaker set at 70 rpm. After 24 h, the medium was switched from a standard expansion medium to an adipogenic differentiation medium (MSCgo™ Adipogenic Differentiation Medium, Sartorius, Göttingen, Germany). The control samples were maintained in a standard expansion medium. Cultures were maintained for 21 days with half of the medium replaced twice per week.

### 2.15. Intracellular Lipid Droplet Staining

Following 21 days of differentiation within *to*-CNF hydrogel spheres, adMSCs were stained with Nile Red (lipid vacuole staining). Samples were fixed with 4% paraformaldehyde for 1 h, washed twice with PBS, and then stained with Nile Red (5 μg/mL in DMSO) for 1 h on a gyroscopic shaker (70 rpm) at 37 °C in the dark. After incubation, samples were washed twice with PBS and fresh PBS was added for fluorescence microscopic analysis. The quantification of the adipogenic differentiation capacity was based on the Nile Red fluorescence-stained area of the images, utilizing ImageJ 1.53t image processing software.

### 2.16. Mechanical Characterization of to-CNF Hydrogels

Compression testing was performed using a miniature material testing frame originally designed for in situ synchrotron ultrasonic fatigue testing [39], as depicted in Figure 1D. The apparatus featured a 50 N capacity load cell and was equipped with two synchronously stepper motor-driven ball screws, providing a 0.3 µm step resolution for crosshead displacement. Data acquisition and stage control were managed by custom software developed in National Instruments LabVIEW 2023 Q3, facilitating precise control over displacement, load, and extension.

Cylindrical hydrogel samples, nominally 5 mm in diameter and 3 mm in height, were transferred into a PEEK cup, submerged in PBS, and subjected to monotonically increasing compression stresses at a rate of 15 µm/s (approximately equivalent to a strain rate of ε = 5 × 10^−3^ s^−1^) until reaching a final length of l = 250 µm. Raw data were collected at a resolution of 16 bits within a nominal 50 N range, sampled at a rate of 10 s^−1^. These raw data were then corrected for load offset, stage compliance, and the initial length l_0_. The initial length l_0_ was estimated as the intersection point between the load offset and the linear least-squares fit of the compression slope within the range of 0 < ε < 0.5, calculated for each sample individually.

### 2.17. Statistical Analysis

The results were expressed as the mean ± standard deviation of at least three independent replicates. Statistical comparisons were performed using unpaired *t*-tests, 1-way ANOVA, and 2-way ANOVA tests followed by Tukey’s multiple comparisons, and data were plotted using GraphPad Prism 8.0.1 for Windows (GraphPad Software, San Diego, CA, USA). Significance was indicated as follows: * *p* < 0.05, ** *p* < 0.01, *** *p* < 0.001, and **** *p* < 0.0001.

## 3. Results

### 3.1. Effect of CaCl_2_-Induced Crosslinking on Mechanical Properties of to-CNF Hydrogel Spheres and Growth Dynamics of Encapsulated adMSCs

Following the preparation of acellular *to*-CNF spheres, both with and without additional CaCl_2_-induced crosslinking (Figure 1A), their stability over a 14-day cultivation period under dynamic conditions was evaluated by observing changes in weight. During this period, no overtime weight changes were detected in either of the examined conditions. However, the CaCl_2_-crosslinked hydrogel spheres exhibited a significantly higher weight compared to those without the additional crosslinking (Figure 1B).

To assess the influence of CaCl_2_-induced crosslinking on the mechanical properties of the hydrogels, a compression test was conducted to determine the elastic moduli of the scaffolds (Figure 1C). Also, the height of the samples was measured (Figure 1D) to further elucidate the potential impact of the CaCl_2_ crosslinking on the swelling of the hydrogels. The results revealed that the addition of CaCl_2_ produced hydrogels with an average height approximately 10% greater and a Young’s modulus roughly 50% lower compared to those formed by self-assembly alone.

In addition to examining the mechanical properties and stability of the *to*-CNF hydrogel spheres, the impact of CaCl_2_ crosslinking on biological systems was assessed over time in statically cultured, cell-laden samples. Specifically, the study examined the effect of CaCl_2_ crosslinking on the viability, proliferation, metabolic activity, and protein secretion of *to*-CNF hydrogel sphere-encapsulated adMSCs. As illustrated in Figure 2A, a higher number of spindle-shaped cells were observed in the hydrogel spheres without CaCl_2_ crosslinking as early as day 7. By day 14, the *to*-CNF hydrogel spheres without CaCl_2_ crosslinking were nearly confluent with spindle-shaped adMSCs, while *to*-CNF hydrogel spheres with CaCl_2_ crosslinking showed only a slight increase in the number of attached and spread cells. Although a small number of dead cells were observed in both culture conditions on day 1, their viability appeared to improve over time, as indicated by a reduction in the observed count of dead cells by day 7. By the end of the cultivation period, some dead cells were still visible, particularly in the overconfluent regions of the *to*-CNF hydrogel spheres without CaCl_2_ crosslinking. However, this observation was qualitative and reflects cell survival challenges during the encapsulation process, and statistical significance was not assessed. A continuous increase in both metabolic activity and cell number was detected in the *to*-CNF hydrogel spheres without CaCl_2_ crosslinking, compared to those crosslinked with CaCl_2_, where cell growth appeared to be impaired. Furthermore, the analysis indicated significantly higher metabolic activity after 14 days (Figure 2B) and a higher number of cells after 7 and 14 days in the samples not crosslinked with CaCl_2_ (Figure 2C). Analysis of the supernatant media protein concentration revealed significantly lower protein levels for the hydrogel spheres additionally crosslinked by CaCl_2_ (Figure 2D).

In summary, the results of this comparative analysis demonstrate that CaCl_2_ crosslinking in addition to gelation by self-assembly significantly impairs both the mechanical properties of the *to*-CNF hydrogel spheres and the growth of encapsulated adMSCs. As a result, all subsequent experiments were conducted using *to*-CNF hydrogel spheres without CaCl_2_ crosslinking, as this condition proved to be more advantageous for the function of the encapsulated cells.

### 3.2. Dynamic Cultivation Enhances adMSC Proliferation and Protein Secretion in to-CNF Hydrogel Spheres

To assess the responsiveness of adMSCs to dynamic cultivation compared to static conditions, cell-laden *to*-CNF hydrogel spheres were prepared according to the methodology outlined in Section 2.3. These spheres were then subjected to either static culture or placed on a gyroscopic shaker to induce dynamic conditions.

Calcein staining did not reveal any notable differences between the two cultivation conditions (Figure 3A). However, analysis of cell proliferation revealed a significantly higher cell count in the *to*-CNF hydrogel spheres cultured under dynamic conditions after 14 days (Figure 3C). Additionally, protein levels in the supernatant medium were significantly higher in the dynamic condition compared to static culture (Figure 3D). While there was a trend towards higher metabolic activity in the dynamic condition, this difference did not reach statistical significance after 14 days of cultivation (Figure 3B).

These results suggest that dynamic cultivation conditions may enhance cell proliferation and protein secretion in CNF hydrogel sphere-encapsulated adMSCs.

### 3.3. Hypoxia Enhances adMSC Proliferation and Protein Secretion in to-CNF Hydrogel Spheres

To evaluate the influence of oxygen tension on adMSCs encapsulated in *to*-CNF hydrogel spheres, prepared as described in Section 2.3, dynamic cultivation was performed under both hypoxic (5% O_2_) and normoxic (21% O_2_) conditions. After 14 days, an increased number of spindle-shaped cells was observed in the *to*-CNF hydrogel spheres cultured under hypoxia, indicative of enhanced cell proliferation and spreading (Figure 4A). Furthermore, the metabolic activity (Figure 4B) and cell number (Figure 4C) were significantly higher in the hypoxia-cultured hydrogel spheres, as evidenced by metabolic assays and DNA content-based cell counting. Additionally, the supernatant protein concentration was also higher in the hypoxic condition (Figure 4D), suggesting increased protein secretion by the adMSCs.

These findings underscore the beneficial effect of hypoxic cultivation conditions on the in vitro expansion of *to*-CNF hydrogel sphere-encapsulated MSCs, which may enhance their diagnostic and therapeutic potential.

### 3.4. Enhanced Adipogenic Differentiation of adMSCs in to-CNF Hydrogel Spheres Under Hypoxic Conditions

The adipogenic differentiation capacity of adMSCs encapsulated in *to*-CNF hydrogel spheres was evaluated in dynamically cultured hypoxic (5% O_2_) and normoxic (21% O_2_) conditions over 21 days. Differentiated cells exhibited a round morphology, contrasting with the spindle-shaped appearance of undifferentiated controls that were overconfluent (Figure 5A). Adipogenic differentiation was quantified through fluorescence image analysis (Figure 5B), revealing a significantly higher quantity of lipid vacuoles in the images from cultures differentiated under hypoxia (Figure 5C).

This observation implies that adMSCs encapsulated in *to*-CNF hydrogel spheres display a small yet significantly increased propensity for adipogenic differentiation when cultured under hypoxic conditions.

### 3.5. Generation and Characterization of Immunomodulatory EVs by to-CNF Hydrogel Sphere-Encapsulated adMSCs

In order to assess the scalability of *to*-CNF hydrogel spheres for adMSC expansion and their potential as a platform for therapeutic EV production, respective samples were cultured in commercially available vertical wheel bioreactors (Figure 6A) under normoxic (21% O_2_) and hypoxic (5% O_2_) conditions. After a 21-day incubation period, there were no discernible differences in terms of the number and spreading of viable cells in both conditions (Figure 6B). In line with the findings in Section 3.3, a significantly higher protein concentration was detected under hypoxic conditions compared to normoxic conditions (Figure 6C). During media changes, the supernatant was collected and stored. At the end of the cultivation period, the collected supernatants were pooled together and processed for EV isolation. NTA conducted on both the total number of particles and EVs alone showed no notable differences between the normoxic and hypoxic conditions in either analysis (Figure 6D). EVs constituted approximately 20% of the total particle count. Following the method suggested by Webber and Clayton [40], when comparing vesicle solution purities, no significant differences were observed between hypoxic and normoxic cultivation in the bioreactor setup (Figure 6E). Additionally, there were no significant differences between normoxic and hypoxic conditions with respect to the average size and size distribution of measured nanoparticles for both the total particles (Figure 6F) and EVs only (Figure 6G).

The immunoregulatory potential of EVs was evaluated using a T-cell-based potency assay. For this, 5 × 10^5^ PBMCs were co-cultured with either 1 × 10^5^ adMSCs or 4 × 10^8^ adMSC-EVs, derived from normoxic and hypoxic cultures in the vertical wheel bioreactor. The concentration of adMSC-EVs used in the PBMC culture was determined based on previous studies [37,38]. The analysis revealed a significant immunosuppressive effect of both normoxic and hypoxic EVs when compared to the blank control. However, no significant discrepancy was observed between the immunosuppressive effects of EVs derived from normoxic and hypoxic conditions. Furthermore, the immunosuppressive capability of adMSCs was significantly higher than that of adMSC EVs (Figure 7A).

EV marker analysis, conducted via microarray assay, unveiled notable differences; specifically, the expression levels of the transmembrane markers CD63 and CD81 were significantly elevated in EVs derived from hypoxic conditions compared to those from normoxic conditions. Conversely, no significant disparity was observed in the expression of the CD9 marker between the normoxic and hypoxic conditions (Figure 7B).

## 4. Discussion

MSCs have immense potential for various clinical applications due to their regenerative and immunomodulatory properties, rendering them promising candidates for a wide array of therapeutic applications [3]. Hydrogels based on nanofibrillar cellulose have emerged as compelling 3D cultivation scaffolds for MSCs, fostering their expansion [41] and differentiation [42] in vitro. The present study delves into the utilization of hydrogel spheres prepared from TEMPO-oxidized and subsequently nanofibrillated cellulose as a novel strategy to augment the in vitro cultivation of adMSCs, with a particular focus on enhancing their capacity to produce EVs as therapeutic agents.

The authors first explored the effects of an additional CaCl_2_-induced crosslinking, alongside the previously described self-assembly of aqueous dilute dispersions of nematically oriented *to*-CNF [15], on the stability and mechanical properties of the hydrogel spheres, as well as their influence on adMSC growth dynamics. The findings indicate that while both *to*-CNF hydrogel sphere formulations remained stable when exposed to flow shear stresses during dynamic cultivation conditions, the introduction of CaCl_2_ crosslinking significantly compromised both the mechanical and biological properties of the hydrogels. In a previous study using the same material, the TEMPO-oxidized cellulose nanofibers were analyzed by conductometric titration and found to have a carboxyl group content of 1.3 mmol/g, corresponding to a degree of oxidation (DO) of 21%. AFM analysis further revealed an average *to*-CNF length of 724 ± 256 nm and a width of 3.3 ± 0.9 nm [15]. Contrary to expectations based on previous studies involving CaCl_2_ crosslinked cellulose hydrogels where such crosslinking was anticipated to enhance mechanical properties [32] without detrimentally affecting cell growth and spreading [33], the present results reveal a different outcome. The authors hypothesize that the surplus of Ca^2+^ ions, introduced by the addition of CaCl_2_, reacted with hydrogen phosphate ions present in the cell culture medium (for MEM alpha: [NaH_2_PO_4_-H_2_O] =140 mg/L, pH = 7) to form poorly soluble calcium phosphates. The latter are likely CaHPO_4_, as HPO_4_^2−^ and H_2_PO_4_^−^ are present in fairly equimolar concentrations at pH 7. Only CaHPO_4_ is poorly water soluble (<0.1 g/L), while the solubility of Ca(H_2_PO_4_)_2_ in H_2_O is about 20 g/L at room temperature (according to the GESTIS substance database). This reaction likely contributed to the observed white coloration and increased opaqueness and weight of the *to*-CNF hydrogel spheres [43]. Several studies suggest that tertiary calcium phosphates, such as Ca_3_(PO_4_)_2_, can precipitate at a higher pH to form crystal lattices inside hydrogels, thereby enhancing their mechanical properties and biostability [44]. The findings on calcium phosphate-modified *to*-CNF hydrogel spheres, however, report a decrease in the elastic modulus and, consequently, mechanical properties due to the deposition of amorphous and likely heterogeneous phosphate particles. Another reason for decreased hydrogel mechanical properties could be the acidification caused by the addition of CaCl_2_. This can not only lead to enhanced hydrogen bonding but also to the formation of CO_2_ from the bicarbonate (NaHCO₃) buffer used in the culture medium. The formed microbubbles are assumed to remain partially entrapped within the *to*-CNF hydrogels, which may account for the slight swelling and reduced mechanical performance of these samples compared to the hydrogels not subjected to additional CaCl_2_ treatment. These insights underscore the critical importance of refining crosslinking strategies to strike a delicate balance between maintaining structural integrity and facilitating favorable cellular interactions.

The authors demonstrated that dynamic cultivation conditions, replicating the perpetual motion encountered by cells in their natural environment, enhance the proliferation of adMSCs and promote protein secretion within *to*-CNF hydrogel spheres. The dynamic culture system may promote nutrient and oxygen transport within the hydrogel matrix, facilitating adMSC growth and metabolism [45]. The observed increase in cellular activity under dynamic cultivation conditions highlights the importance of simultaneous exposure to external forces, material properties, and cellular microenvironment characteristics in fostering cell growth and functionality, as elucidated in the existing literature [46,47]. These findings align with prior research illustrating the advantages of incorporating dynamic culture systems alongside hydrogel scaffolds to enhance cell proliferation [48] and protein secretion [49]. Despite being recommended, there has been limited exploration into the influence of combining multiple mechanical signals (e.g., hydrogel stiffness, flow shear stress, surface topography, cyclic strain, etc.) on the proper functioning of cells in vitro [47].

Hypoxic culture conditions have emerged as a critical factor influencing MSC behavior and function [23]. The present results demonstrate that hypoxic cultivation significantly enhances adMSC proliferation and protein secretion within *to*-CNF hydrogel spheres. Furthermore, the hypoxic environment appears to create an optimal microenvironment that promotes the MSCs’ commitment towards adipogenic lineage, potentially enhancing the regenerative capacity of these cells in therapeutic applications. These findings are consistent with previous studies showing that hypoxia promotes MSC proliferation [50], differentiation [51], and immunomodulation [52] by activating hypoxia-inducible factors (HIFs) and downstream signaling pathways. Moreover, hypoxic culture conditions have been shown to enhance EV production and functionality [53,54], further highlighting the potential therapeutic benefits of hypoxic culture for MSC-based therapies [24,55].

This study also investigated the production of EVs by adMSCs encapsulated within *to*-CNF hydrogel spheres in a scalable bioreactor system. While bioreactor systems have been employed for MSC expansion, their effects on EV production and functionality remain subjects of ongoing research. Transferring cell growth conditions from a conventional 2D static culture to advanced 3D cultivation systems necessitates adaptation and optimization. Nonetheless, opting for 3D bioreactor systems offers advantages such as improved control and monitoring of culture parameters, scalability, and provision of more physiologically relevant conditions. Given the existing discrepancies [56] in the literature concerning quantitation methods for EV production, no standardized procedures for the isolation and purification of MSC EVs have been established [35]. Emerging evidence suggests that many particles in various EV preparations may not actually be EVs [35]. Furthermore, recently discovered nonvesicular extracellular nanoparticles, such as exomeres and supermeres, contain many of the biomolecules—including proteins, RNA, and DNA—that were previously attributed to exosomes [57]. This presents a significant challenge due to the heterogeneity of EVs and nonvesicular extracellular nanoparticles and the complexities involved in isolating and purifying distinct populations [57]. Therefore, the authors opted to present their findings in terms of the total number of pure EVs, alongside their purity ratio relative to protein content [40] and the total nanoparticle count. This approach facilitates comparisons with both current and forthcoming study outcomes. While no significant differences were observed in EV production between the normoxic and hypoxic conditions, both conditions exhibited potent immunosuppressive properties. Moreover, observations regarding the immunosuppressive efficacy of adMSCs and adMSC EVs align with findings from previous studies utilizing the same T-cell-based immunopotency assay [36,38]. These findings suggest that *to*-CNF hydrogel spheres may serve as an effective platform for the scalable production of therapeutic EVs for immunomodulatory therapies. Additionally, microarray analysis revealed differences in EV surface marker expression between normoxic and hypoxic conditions, with hypoxic conditions showing an elevated expression of CD63 and CD81 markers. These biomarkers, established by the International Society for Extracellular Vesicles as minimal biophysical standards, are pivotal for defining EVs and assessing their functionality. Although the precise functional roles of these tetraspanins in EVs remain unclear, and no studies have directly linked their expression to oxygen tension and immunomodulation, a few studies have associated their expression with EV biogenesis [58,59] and with binding and uptake by target cells [60,61]. These findings suggest that hypoxic conditions during culture may enhance the generation and therapeutic efficacy of MSC-derived EVs by influencing mechanisms other than immunomodulation. However, further studies are warranted to elucidate the underlying mechanisms and validate the therapeutic potential of hypoxic EVs in preclinical models.

## 5. Conclusions

Overall, these findings underscore the importance of optimizing fundamental culture conditions and hydrogel properties for enhancing the therapeutic potential of MSCs. *to*-CNF hydrogel spheres provide a versatile platform for both MSC cultivation and EV production. These hydrogel spheres emulate an ECM-like environment, promoting favorable cellular responses, particularly under more physiological cultivation conditions. This versatility holds promise for applications spanning tissue engineering, regenerative medicine, and immunomodulatory therapies. Future studies should further explore the mechanisms underlying the effects of culture conditions on MSC behavior and function, as well as the therapeutic potential of EVs produced by the MSCs in *to*-CNF hydrogel spheres.

## Figures and Tables

**Figure 1 bioengineering-11-01050-f001:**
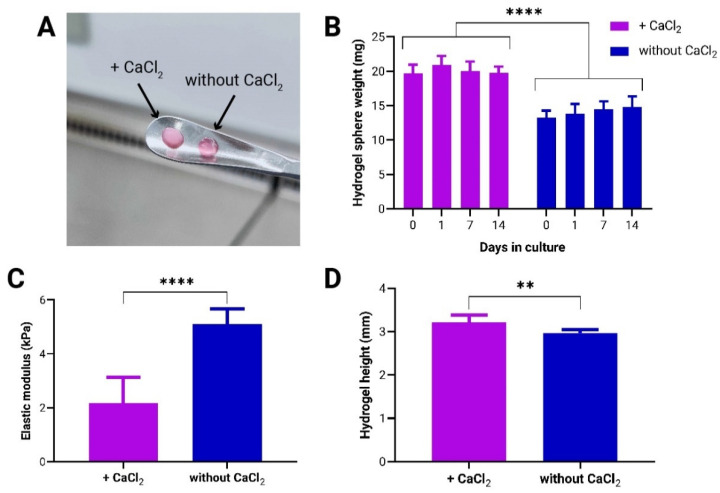
Mechanical stability and characterization of *to*-CNF hydrogel spheres with and without additional CaCl_2_-induced crosslinking. (**A**) *to*-CNF hydrogel spheres with (left) and without (right) CaCl_2_ crosslinking. (**B**) Stability assessment of hydrogel spheres over 14 days under dynamic conditions (n = 6, 2-way ANOVA, *t*-test). (**C**) Elastic modulus measurements of *to*-CNF hydrogels with and without CaCl_2_ crosslinking (n = 8, *t*-test). (**D**) Measurement of hydrogel sample height, prepared with and without CaCl_2_ crosslinking, for compression testing (n = 8, *t*-test).

**Figure 2 bioengineering-11-01050-f002:**
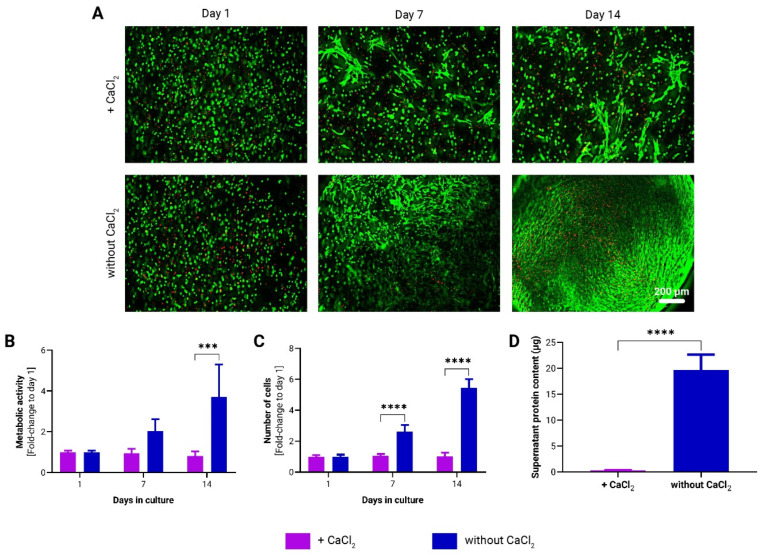
Growth dynamics of adMSCs encapsulated in *to*-CNF hydrogel spheres with and without CaCl_2_ crosslinking. (**A**) Live/dead staining of encapsulated adMSCs at 1, 7, and 14 days post-encapsulation in *to*-CNF hydrogel spheres. Corresponding changes in metabolic activity (**B**) and cell number (**C**) over time in the *to*-CNF hydrogel spheres (n = 3, 2-way ANOVA). (**D**) Analysis of supernatant protein concentration (n = 3, *t*-test).

**Figure 3 bioengineering-11-01050-f003:**
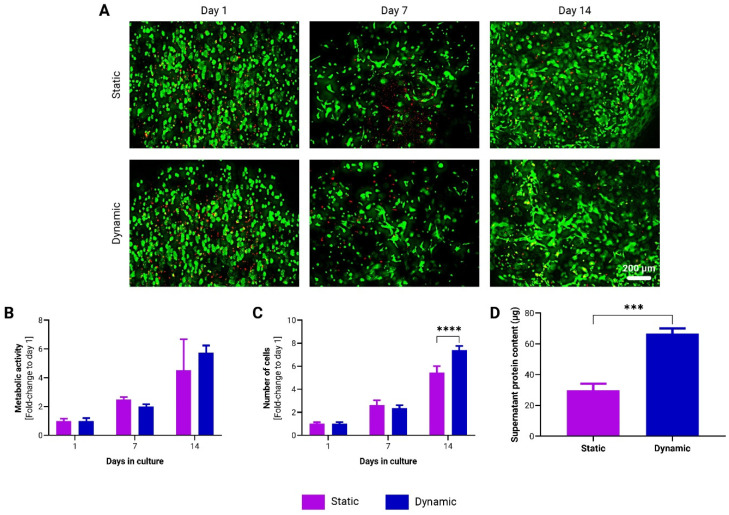
Growth kinetics of adMSCs encapsulated in CNF hydrogel spheres without CaCl_2_ crosslinking, under static and dynamic cultivation conditions. (**A**) Representative images of live/dead staining of encapsulated adMSCs after 1, 7, and 14 days following encapsulation in CNF hydrogel spheres. Over time changes in the metabolic activity (**B**) and in the number of adMSCs (**C**) encapsulated in CNF hydrogel spheres. (n = 3, 2-way ANOVA). (**D**) Analysis of supernatant protein concentration (n = 3, *t*-test).

**Figure 4 bioengineering-11-01050-f004:**
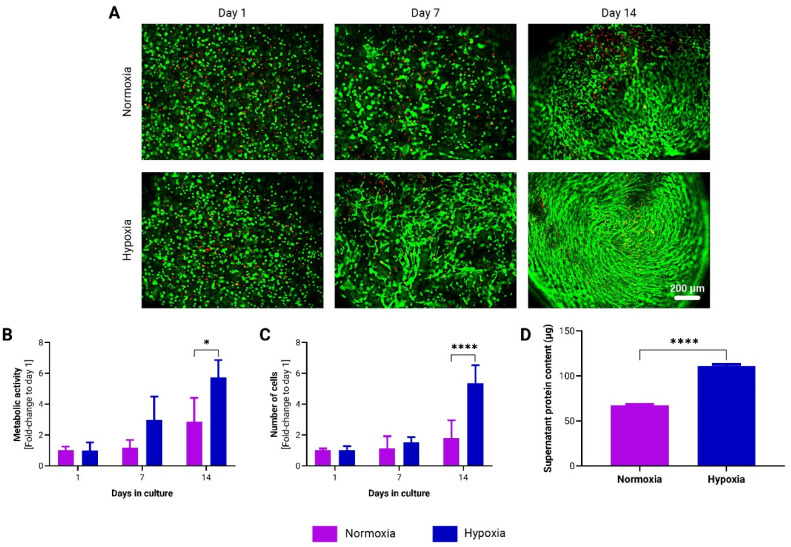
Growth dynamics of adMSCs encapsulated in *to*-CNF hydrogel spheres without CaCl_2_ crosslinking, and cultivated under normoxic (21% O_2_) and hypoxic (5% O_2_) dynamic conditions. (**A**) Representative images of live/dead staining of encapsulated adMSCs after 1, 7, and 14 days following encapsulation in *to*-CNF hydrogel spheres. Changes in the metabolic activity (**B**) and in the number of adMSCs (**C**) encapsulated in *to*-CNF hydrogel spheres (n = 3, 2-way ANOVA). (**D**) Analysis of supernatant protein concentration (n = 3, *t*-test).

**Figure 5 bioengineering-11-01050-f005:**
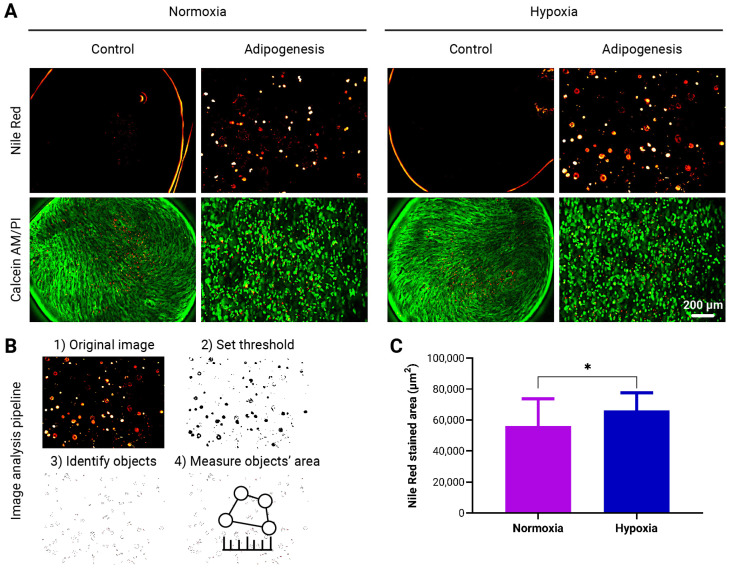
Evaluation of adMSCs Adipogenic Differentiation within *to*-CNF hydrogel spheres without CaCl_2_ crosslinking. (**A**) Representative images showing lipid vacuole formation and live/dead staining of encapsulated adMSCs after 21 days of adipogenic differentiation. (**B**) Image analysis pipeline using ImageJ software to quantify the fluorescence-stained area of the images. (**C**) Quantification of the adipogenic differentiation capacity of the *to*-CNF hydrogel sphere-encapsulated cells under normoxia and hypoxia, based on image analysis data (n = 20, *t*-test).

**Figure 6 bioengineering-11-01050-f006:**
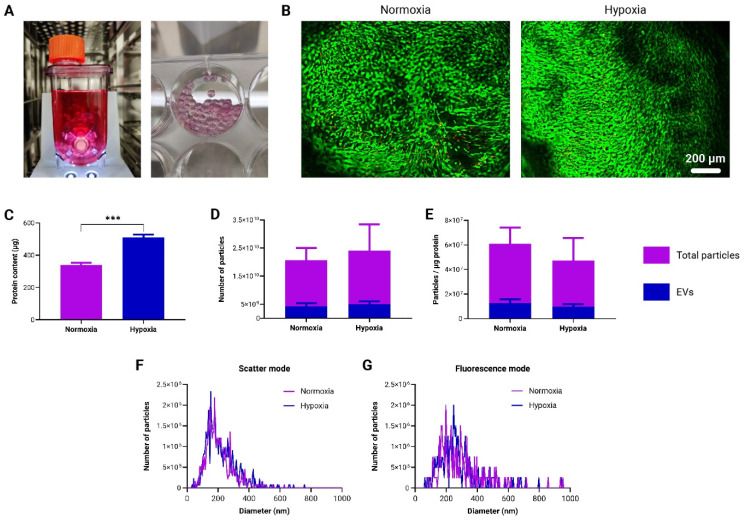
Production and characterization of EVs derived from *to*-CNF hydrogel sphere-encapsulated adMSCs, without CaCl_2_ crosslinking, in the vertical wheel bioreactor. (**A**) Depictions of the vertical wheel bioreactor and *to*-CNF hydrogel spheres post-21 days of cultivation. (**B**) Fluorescence images showing live/dead staining outcomes of encapsulated adMSCs after 21 days of cultivation in the bioreactor within *to*-CNF hydrogel spheres. (**C**) Quantitative analysis of supernatant protein concentration (n = 3, multiple *t*-tests). (**D**) Nanoparticle Tracking Analysis (NTA) results depicting the total particle and EV-only numbers under hypoxia and normoxia (n = 11, multiple *t*-tests). (**E**) Assessment of vesicle solution purity for both “total particles” and “EVs only” under normoxic and hypoxic bioreactor cultivation conditions (n = 11, multiple *t*-tests). NTA results in scatter (total particles) (**F**) and fluorescence (EVs) (**G**) mode, presenting nanoparticle size distribution (Kolmogorov–Smirnov test).

**Figure 7 bioengineering-11-01050-f007:**
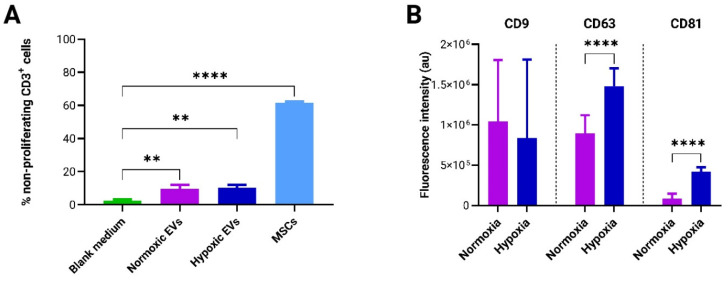
Assessment of the immunomodulatory potential and surface marker analysis of EVs produced in differential culture conditions. (**A**) Evaluation of T-cell proliferation inhibition potency exhibited by both adMSCs and adMSC EVs co-cultured with pooled CFSE pre-labeled PBMCs for a duration of 4 days (n = 3, 1-way ANOVA). (**B**) Microarray-based quantification of EV surface marker expression profiles (n = 9, multiple *t*-tests).

## Data Availability

A part of this study was previously presented as a short talk at the 15th ÖGMBT Annual Meeting 2023 in Salzburg, Austria.

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
