# Peer review of "Characterization of MSC Growth, Differentiation, and EV Production in CNF Hydrogels Under Static and Dynamic Cultures in Hypoxic and Normoxic Conditions"

_bioengineering, 2024, doi:10.3390/bioengineering11101050_

Round 1
Reviewer 1 Report
Comments and Suggestions for Authors
In this work, Nikolits et al. provided the basic characterization Mesenchymal stem cells (MSCs) in (i) static and dynamic culture and (ii) hypoxic and normoxic conditions, laying the foundation for further work. The study is comprehensive. I would recommend the publication.
Author Response
Comment: In this work, Nikolits et al. provided the basic characterization Mesenchymal stem cells (MSCs) in (i) static and dynamic culture and (ii) hypoxic and normoxic conditions, laying the foundation for further work. The study is comprehensive. I would recommend the publication.
Response: Thank you very much for your positive and encouraging feedback on our manuscript. We greatly appreciate your time and consideration in reviewing our work. We are pleased that you found our study comprehensive and hope that it will contribute meaningfully to the ongoing development of MSC-based research and therapeutic applications.
Reviewer 2 Report
Comments and Suggestions for Authors
Comments and Suggestions for Authors
This is an elegant and insightful study regarding the characterisation of MSC Growth, Differentiation, and EV Production in CNF Hydrogels under Static and Dynamic Cultures in Hypoxic and Normoxic Conditions.
The manuscript presents a well-structured study on optimizing in vitro cultivation conditions for mesenchymal stem cells (MSCs) using cellulosic hydrogels. The focus on comparing two physical crosslinking methods and analyzing their impact on MSC proliferation, differentiation, and extracellular vesicle (EV) production is particularly relevant for advancing regenerative medicine and therapeutic applications.
Additionally, the improvements observed under dynamic and hypoxic conditions are noteworthy. Moreover, while the potential for scalable EV production is intriguing, more information on the specific bioreactor system and its scalability could strengthen the practical implications of the study.
This research offers valuable insights into MSC cultivation and highlights key factors for improving their therapeutic potential.
The purpose is well-identified. The work is well outlined, well organized, and relevant. The results support the discussion. Overall, it is a significant study on this matter. However, to be published, in my opinion, there are some minor changes to be made:
1) Page 3, Lines 102-132: In my opinion, the first-person plural (we) should not be used. The authors should revise these paragraphs using "The authors".
2) Page 3, Lines 122-132: In the introduction, it makes sense to state the objective of the study (mentioned in the previous paragraph). The results should be in the conclusion or discussion. At most, a brief reference, but never an entire paragraph."
3) Page 9, Lines 419-421: The sentence “Although a small number of dead cells were observed in both conditions on day 1, their count decreased by day 7. By the end of the cultivation period, some dead cells could be observed in the overconfluent areas of to-CNF hydrogel spheres without CaCl2 cross-linking.” is not clear. It is not possible to understand what the authors mean by this. Please make it clearer, explicitly stating which result you are referring to and whether it is statistically significant.
4) Small format change in the abstract (Line 18-21) should not be it italics.
The results are very interesting and the discussion and conclusion are supported by the results. Very well organized and supported. Therefore, in my opinion, after these changes the manuscript is publishable.
Author Response
We would like to extend our gratitude for your thorough and insightful feedback on our manuscript. Your positive remarks and constructive suggestions are highly appreciated, and we have made the following revisions accordingly:
1. Comment: Page 3, Lines 102-132: In my opinion, the first-person plural (we) should not be used. The authors should revise these paragraphs using "The authors".
Response: We have revised the entire manuscript as per your suggestion, replacing the first-person plural "we" with "the authors" to maintain a more formal and objective tone throughout.
2. Comment: Page 3, Lines 122-132: In the introduction, it makes sense to state the objective of the study (mentioned in the previous paragraph). The results should be in the conclusion or discussion. At most, a brief reference, but never an entire paragraph.
Response: As recommended, we have revised the introduction by removing the detailed description of the results and limited this section to a clear statement of the study's objectives. The relevant results are highlighted in the discussion and conclusion sections, ensuring that the introduction now focuses solely on setting the context and purpose of the study.
3. Comment: Page 9, Lines 419-421: The sentence “Although a small number of dead cells were observed in both conditions on day 1, their count decreased by day 7. By the end of the cultivation period, some dead cells could be observed in the overconfluent areas of to-CNF hydrogel spheres without CaCl2 cross-linking.” is not clear. It is not possible to understand what the authors mean by this. Please make it clearer, explicitly stating which result you are referring to and whether it is statistically significant.
Response: We appreciate your thoughtful comment and agree that the phrasing could be made clearer. We have revised the sentence as follows to improve clarity:
"Although a small number of dead cells were observed in both culture conditions on day 1, their viability appeared to improve over time, as indicated by a reduction in the observed count of dead cells by day 7. By the end of the cultivation period, some dead cells were still visible, particularly in the overconfluent regions of the to-CNF hydrogel spheres without CaCl2 cross-linking. However, this observation was qualitative and reflects cell survival challenges during the encapsulation process, and statistical significance was not assessed."
We would also like to clarify that this observation pertains to qualitative analysis, and as such, statistical significance was not assessed. We thank you for bringing this to our attention and allowing us to clarify this point in the manuscript.
4. Comment: Small format change in the abstract (Line 18-21) should not be in italics.
Response: We apologize for the formatting issue. This change occurred during the transposition from the format-free manuscript to the journal’s template. Thank you for pointing this out, and we will ensure that this formatting error is corrected during the final formatting stage.
Reviewer 3 Report
Comments and Suggestions for Authors
This manuscript investigates mesenchymal stem cell (MSC) behaviors in cellulose nanofiber (CNF) hydrogels under varied culture conditions. It excels in its exploration of dynamic and hypoxic conditions on MSC proliferation and extracellular vesicle (EV) production, offering innovative insights into scaffold design for therapeutic applications.
Here are some comments and suggestions:
1. Please check the font from line 18 to line 21.
2. Additional experiments comparing CNF hydrogels with other hydrogel materials could clarify the specific advantages or limitations of CNF hydrogels for MSC applications.
3. The study would benefit from a detailed correlation between the mechanical properties of the hydrogels and MSC behavior, to better understand the material's clinical suitability.
4. The manuscript presents short-term results but does not provide long-term data on cell viability and functionality, which are crucial for potential clinical applications.
5. While the study mentions the immunomodulatory potential of MSC-derived EVs, it provides limited analysis of the specific immune responses elicited by these vesicles. A more detailed examination of the immune interactions, possibly through additional in vivo studies, would enhance the understanding of the therapeutic potential of these EVs.
Author Response
We would like to express our gratitude for your valuable feedback and constructive comments on our manuscript. Below, we address your comments in detail.
- Comment: Please check the font from line 18 to line 21
Response: We apologize for the formatting issue. This change occurred during the transposition from the format-free manuscript to the journal’s template. Thank you for pointing this out, and we will ensure that this formatting error is corrected during the final formatting stage.
- Comment: Additional experiments comparing CNF hydrogels with other hydrogel materials could clarify the specific advantages or limitations of CNF hydrogels for MSC applications.
Response: We appreciate your suggestion regarding the comparison of CNF hydrogels with other hydrogel materials. While this comparison makes a lot of sense and is indeed our aim for future studies, the current study focuses on exploring the novel application capacities of CNF hydrogels for MSC in vitro applications as a starting point. By concentrating on the unique properties and applications of CNF hydrogels, we hope to lay a solid foundation for subsequent investigations that could include comparative analyses with other materials.
- Comment: The study would benefit from a detailed correlation between the mechanical properties of the hydrogels and MSC behavior, to better understand the material's clinical suitability.
Response: We acknowledge the importance of correlating the mechanical properties of CNF hydrogels with MSC behavior. In our previous publication, we performed such correlations, which can be found at the following link: https://pubs.acs.org/doi/10.1021/acsabm.2c00854. In the present study, our focus was primarily on improving MSC cultivation through physiological conditions and enhancing EV production potential.
- Comment: The manuscript presents short-term results but does not provide long-term data on cell viability and functionality, which are crucial for potential clinical applications.
Response: In our study, we conducted cultivation periods of 21-28 days, which is considered long-term for in vitro applications. While our studied material applications focus on in vitro modeling, drug screening, and EV production, these cultivation times are appropriate for these purposes. However, we acknowledge that clinical in vivo applications would require extended cultivation periods and further evaluation. It is also important to note that the tested material may be limited for in vivo clinical applications due to its lack of biodegradability by mammalian cells.
- Comment: While the study mentions the immunomodulatory potential of MSC-derived EVs, it provides limited analysis of the specific immune responses elicited by these vesicles. A more detailed examination of the immune interactions, possibly through additional in vivo studies, would enhance the understanding of the therapeutic potential of these EVs.
Response: According to recent literature, the typical approach for initial screening of MSC-produced EVs involves evaluating their immunomodulatory capacities, similar to our methods (https://doi.org/10.3727/096368915x687543). There remains considerable discrepancy and a lack of clear standards for further analysis in this field (https://doi.org/10.3390%2Fcells9051171). While in vivo studies would provide additional confirmation of therapeutic potential, establishing robust in vitro evaluation standards is crucial before advancing to in vivo analyses. Our study aligns with similar recent publications in the field, focusing on foundational assessments before embarking on more complex investigations (https://doi.org/10.3390%2Fijms18071413).